# Application of Hardware-in-the-Loop Simulation Technology in the Development of Electro-Hydraulic Servo System Control Algorithms

**Quan Liang** *,†, **Jun Gao** ‡, **Feihong Liu** ‡, **Kelei Wang, Haiyang Zhang, Zhike Wang and Donghai Su**

Mechanical Engineering Department, Shenyang University of Technology, Shenyang 110870, China

* Correspondence: liangquan@sut.edu.cn; Tel.: +86-180-0402-6017
† Current address: Shenyang Economic and Technological Development Zone, 111 Shenliao West Road, Shenyang 110870, China.
‡ These authors contributed equally to this work.

**Abstract:** In this paper, we present a method to identify paramaters for controlling electro-hydraulic servo systems in a real-time environment. With the aim of addressing the problem that it is difficult to accurately obtain the state space equation parameters of the physical entity of the electro-hydraulic servo system, we introduce an online identification theory (recursive least squares method) for identifying said parameters of the state space model in a valve-controlled symmetrical cylinder system. After accurately obtaining the parameters of the system, nonlinear control of the valve-controlled symmetrical cylinder system is carried out using a backstepping algorithm. In order to verify the actual effect of the online identification algorithm and backstepping algorithm, a hardware-in-the-loop (HIL) simulation platform for the valve-controlled symmetrical cylinder system is built in a Linux real-time system, and the real-time performance of the system is evaluated, which demonstrates that the platform can be reliably applied for subsequent system identification and backstepping verification. The results of the HIL simulation test demonstrate that the online identification algorithm and backstepping control method developed in this paper are effective and reliable.

**Keywords:** electro-hydraulic servo system; system identification; backstepping control; hardware-in-the-loop simulation

## 1. Introduction

Electro-hydraulic servo (EHS) systems and EHS actuators form one of the fundamental drive systems applied in industrial processes and engineering practices. Compared with electrical servo systems, EHS systems typically have higher power to weight ratios, fast and smooth start-up characteristics, and higher power storage capabilities [1,2]

The ability of EHS actuators to perform their powerful control functions is heavily dependent on the control algorithms. Typically, EHS systems, or the mechanisms they drive, exhibit strong nonlinearity [3–6]. Thus, nonlinear control algorithms are particularly suitable for EHS actuators [7–14], of which the performance of the backstepping method is the most prominent [15–19]. However, the backstepping method is highly dependent on the dynamic model of the controlled system and its parameters, which are usually difficult to obtain accurately and requires the use of adaptive control algorithms [20–28] or identification methods to obtain the exact parameters of the system [14].

In the past few decades, numerous scholars have carried out research work focused on parameter identification and nonlinear control algorithms for EHS systems. In [29], some practical issues concerning the identification of EHS actuators are discussed, with the use of discrete time linear models. Hydraulic wind power transfer systems exhibit a highly nonlinear dynamic influenced by system actuator hysteresis and disturbances from wind

speed and load torque. A system identification approach to approximate such a nonlinear dynamic is presented in [30]. Wos [31] presents selected issues of position force control of EHS system using adaptive methods. This kind of measure extends the capabilities of control systems that only use position measurements. In [32], experimental work is presented on nonrecursive identification of an EHS actuator system that is represented by a discrete time model in open-loop configuration. A least squares method is used to estimate the unknown parameters of the system based on autoregression with an exogenous input (ARX) model. Jin [33] proposes a novel double-layered network scheme for accurate and efficient identification of hydraulic cylinder. Paper [34,35] presents experimental online identification of an electro-hydraulic system represented by a discrete time model. A recursive least squares (RLS) method is used to estimate the unknown parameters of the system based on autoregression using an exogenous input (ARX) model. Paper [36] presents the system identification process conducted on an industrial EHS actuator system. Ghazali [37] presents system identification of EHS actuator system using a nonrecursive or offline technique, where the mathematical model is determined first by considering fixed supply pressure and load. Lee [38] proposed a system identification algorithm of an EHS for a feedforward position control method. Cologni [39] proposed a global parametric identification procedure that has been performed to identify all unknown parameters.

The common point of the above studies is that, on the one hand, the model parameters of the electro-hydraulic servo system are obtained by means of the system identification algorithm; on the other hand, the parameters obtained by the system identification are applied to the control algorithm in order to obtain improved performance.

However, since both the system identification algorithm and the backstepping control method are complex, debugging the algorithm directly on the physical entity of the controlled object is risky and inefficient. Therefore, this paper makes use of hardware-in-the-loop (HIL) simulation technology to debug the system identification algorithm and the backstepping control algorithm in order to remedy the problem of the difficult development and debugging of control algorithms. Past studies on HIL simulation of EHS have focused on electro-hydraulic braking systems [40–46]. In addition, there are examples in the literature of HIL simulation for testing of electro-hydraulic fuel control unit in a jet engine application [47]. However, none of the aforementioned articles describes the method of building a HIL simulation platform in a Linux real-time environment, nor do they describe the debugging of system identification and control algorithms with the help of a HIL simulation platform. In this paper, HIL simulation technology is applied to the development and debugging of an online identification algorithm and backstepping control algorithm, which reduces the difficulty and risk associated with programs development and improves the efficiency.

The paper then introduces the basic principle of system identification and the control algorithm combined with nonlinear backstepping control and the method for its implementation; then, the nonlinear backstepping control algorithm for valve-controlled symmetric cylinder (VCSC) is derived; finally, the method of building the HIL simulation platform based on a Linux real-time system is introduced, and the identification and backstepping control method is carried out using this method. Finally, the HIL simulation platform based on Linux real-time system is presented and used to conduct identification and backstepping control experiments.

## 2. Electro-Hydraulic Servo System

The HIL simulation technique is reliant on the accurate mathematical description of the system dynamic characteristics model. The object of HIL simulation in this paper is an EHS system. Therefore, we first introduce the method for derivation of the mathematical model of the EHS system.

The typical actuator of the EHS system is the servo hydraulic cylinder or motor, where the former mainly executes the linear motion, and the latter mainly executes the rotary

servo motion. Compared with a linear electrical motor, the former has higher power density ratio and faster response speed, so it is irreplaceable in some instances.

From a representative point of view, a valve-controlled double piston rod hydraulic cylinder is selected as a typical model for introduction, and the dynamic model of the valve-controlled hydraulic motor and cylinder is similar. In this study, it is assumed that the EHS cylinder is vertically mounted with the inertial load placed at one end of the piston rod, and a schematic diagram is shown in Figure 1. This EHS system mainly consists of a servo valve and a hydraulic cylinder and also includes pressure sensors for measuring displacement, external load, and two chambers of the hydraulic cylinder. The data obtained from these sensors will be used in the system identification and backstepping control algorithm presented in this paper.

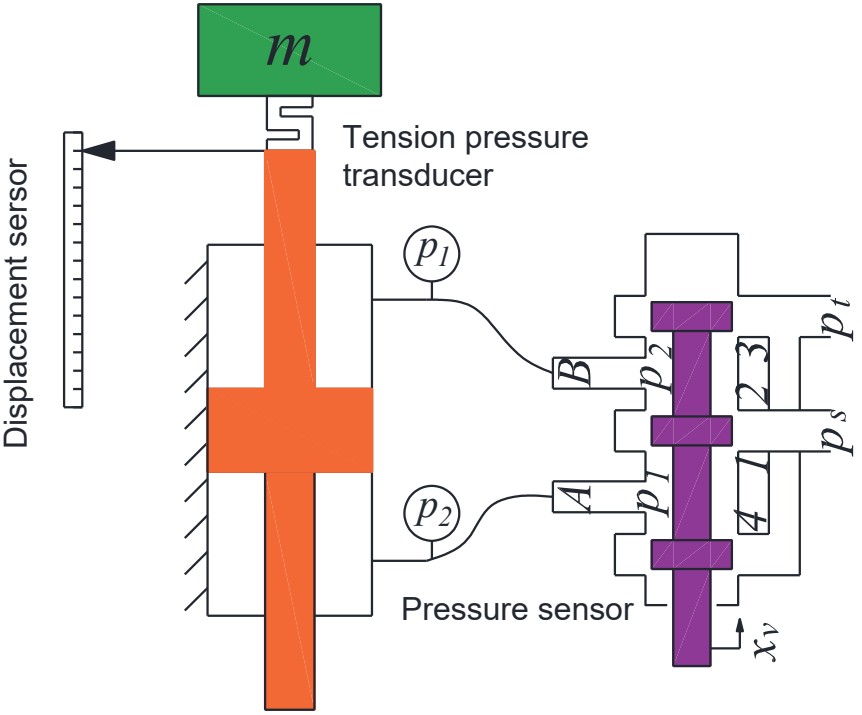

**Figure 1.** Schematic diagram of electro-hydraulic servo system for VCSC.

### 2.1. Servo Valve

A servo valve is the most important control element in the EHS system, which amplifies the electrical signal and converts it into high power density hydraulic energy to drive the actuator to externally perform the work. The dynamic characteristics of the servo valve largely affect the overall performance of the EHS system, for which it is usually difficult to build a detailed mathematical model. For simplicity, the model of servo valve in this study is considered as first-order system, that is, the spool valve displacement $x_v$ is related to the voltage input $u$ by a first-order system given by

$$\dot{x}_v = -\frac{1}{\tau_v}x_v + \frac{k_v}{\tau_v}u \tag{1}$$

where $\tau_v$ and $k_v$ are the time constant and gain of the servo valve respectively.

### 2.2. Load Model

In this paper, it is assumed that the EHS cylinder drives the inertial load to complete the motion, and the EHS cylinder and its load are vertically arranged. This way, the system will always be subjected to an external force in the vertical direction equal to the gravitational force applied to the inertial load, so it is necessary to establish the differential

equation of the dynamic characteristics of the inertial load. According to Newton's second law, the dynamics of the inertia load can be described by [48]

$$m\ddot{x} = p_L A_c - b_c \dot{x} - k_c x - F_L \tag{2}$$

where $x$ and $m$ represent the displacement and the mass of the load, respectively. $p_L = p_1 - p_2$ is the load pressure of the cylinder, $p_1$ and $p_2$ are the left and right chamber pressure, respectively, $A_c$ is the active area of the cylinder, and $b_c$ represents the combined coefficient of the modeled damping and viscous friction forces on the load and the cylinder rod. $k_c$ is the spring stiffness of the external load, which in this case is mainly determined by the elastic deformation of the tension pressure sensor and the spring stiffness of the attached mechanical components. $F_L$ is the external load force. In this paper, since the hydraulic cylinder is assumed to be vertically arranged, the external load force is gravity, that is, $F_L = mg$, where $g$ is the acceleration due to gravity.

### 2.3. Actuator

Not ignoring the effect of leakage flows in the cylinder and servo valve, the actuator (or the cylinder) dynamics can be written as [48,49]

$$\frac{V_t}{4\beta_e} \dot{p}_L = -A_c \dot{x} - C_l p_L + Q_L \tag{3}$$

where $V_t$ is the total volume of the two-chamber cylinder and hoses between it and the servo valve, $\beta_e$ is the effective bulk modulus, $C_l$ is the coefficient of the total internal leakage of the cylinder due to pressure, and $Q_L$ is the load flow. $Q_L$ is related to the spool valve displacement of the servo valve by

$$Q_L = C_d w x_v \sqrt{\frac{p_s - \text{sgn}(x_v) p_L}{\rho}} \tag{4}$$

where $C_d$ is the discharge coefficient, $w$ is the spool valve area gradient for cylindrical slide valves, $w = \pi d_v$, $d_v$ is the spool diameter, and $p_s$ is the supply pressure of the system.

In the load flow formula, the function $\text{sgn}(*)$ is used to distinguish the positive and negative directions of the hydraulic cylinder movement sign function. Its value is

$$\text{sgn}(*) = \begin{cases} 1, * \geq 0 \\ -1, * < 0 \end{cases} \tag{5}$$

However, the existence of this function causes the load flow formula to be discontinuous, which will cause the mathematical problem of derivative discontinuity, whether for system identification or nonlinear control. Therefore, without loss of generality, the above discontinuous function is replaced by the hyperbolic tangent [49], which is defined as

$$\text{sgn}(*) = \tanh(k_a *) \tag{6}$$

where $k_a$ is a proportional coefficient, which absolutely determines the slope of the hyperbolic tangent function.

Define the state variables $x = [x_1, x_2, x_3, x_4]^T \triangleq [x, \dot{x}, p_L, x_v]^T$. The system can be expressed in state space form as

$$
\begin{aligned}
\dot{x}_1 &= x_2 \\
\dot{x}_2 &= -\frac{k_c}{m} x_1 - \frac{b_c}{m} x_2 + \frac{A_c}{m} x_3 - \frac{1}{m} F_L \\
\dot{x}_3 &= -\frac{4\beta_e A_c}{V_t} x_2 - \frac{4\beta_e C_l}{V_t} x_3 + \frac{4\beta_e}{V_t} C_d w \sqrt{\frac{1}{\rho}} \sqrt{|p_s - \tanh(k_a x_4) x_3|} x_4 \\
\dot{x}_4 &= -\frac{1}{\tau_v} + \frac{k_v}{\tau_v} u \\
y &= x_1
\end{aligned}
\tag{7}
$$

The above mathematical model is characterized by the use of function tanh instead of function sgn such that the discontinuous mathematical model is transformed into a continuous model. The physical characteristics of the valve-controlled cylinder system are correctly simulated, which provides the precondition for subsequent development of the HIL simulation system.

## 3. System Identification

To obtain increased performance of EHS systems, it is usually necessary to accurately obtain the model system parameters. Due to the common EHS actuator operating mainly through the throttle action of the servo valve to achieve servo control, the mathematical relationship between the pressure difference before and after the throttle orifice of the servo valve and the flow through the servo valve is nonlinear (square root). At the same time, the EHS system parameters, such as the elastic modulus of the oil and the flow coefficient of the throttle port, are easily affected by external factors and time-varying, which makes the dynamic model of the system and its key parameters difficult to obtain under normal circumstances.

For some control algorithms, such as PID control, it is not necessary to obtain accurate parameters of the system; for other algorithms, such as the nonlinear backstepping method, accurate parameters of the system are usually required to obtain better control results than with the PID control effect. More importantly, for the ordinary PID algorithm, when the parameters of the controlled system change due to some reason, the control algorithm has no ability to adapt to this change. If the parameters of system can be obtained online in real time, even if some parameters change, the control algorithm can adapt to this and then achieve the best control effect.

Therefore, the EHS system is dependent on the following two factors to obtain a good control effect:

- To be able to obtain accurate mathematical model and parameters of the system;
- if there is a change in parameters over time, it is necessary to obtain the changing parameter values in real time to modify the control algorithm.

Successfully addressing the above two factors depends on the system identification technology.

The purpose of system identification is to calculate the unknown parameters of the system by measuring the input and output data of the dynamic characteristics of the model with the help of mathematical algorithms. System identification is usually divided into offline and online. The advantage of the former is that the algorithm is simple. The disadvantage of the latter is the need to acquire a sufficient set of experiment test data of the system, which may require a long time and large storage space. More importantly,to achieve a better control effect, the parameter values obtained by offline identification can only be applied to the control algorithm after the identification work is completed. The online identification technology can obtain the key parameters of the system online and in real time such that the control algorithm relying on the dynamic characteristics of the system parameters can achieve better control results, such as in the case of the nonlinear method. Online identification saves time in data collection and improves the model accuracy and reliability. Moreover, any changes in the system structure and components will be taken into consideration in the model without the need for data collection.

Online identification can be defined as an algorithm that estimates dynamic mathematical model parameters using data measured in a real-time environment. In this paper, the dynamic model is the estimated state space function of the electro-hydraulic servo system.

The most basic method of system identification is the least squares method. The problem to be solved by least squares is to find $\theta_i, i = 1, \dots, n$ according a series of data for known $x_i, i = 1, \dots, n$ and $y$ , i.e.,

$$\mathbf{y} = [x_1, \dots, x_n] \begin{bmatrix} \theta_1 \\ \vdots \\ \theta_n \end{bmatrix}$$

written as a matrix of the form

$$
\begin{bmatrix} y_1 \\ \vdots \\ y_k \end{bmatrix} = \begin{bmatrix} \phi_1^T \\ \vdots \\ \phi_k^T \end{bmatrix} \boldsymbol{\theta}
\tag{8}
$$

where

$$
\phi_i^T = \begin{bmatrix} x_1^i & \cdots & x_n^i \end{bmatrix} \in \mathbb{R}^{1 \times n}
$$

$$
\boldsymbol{\theta} = \begin{bmatrix} \theta_1 \\ \vdots \\ \theta_n \end{bmatrix} \in \mathbb{R}^{n \times 1}
$$

The right-hand side of the Equation (8) represents input observations for the $i$th set of data, and the left-hand side represents the output observations for the $i$th set of data, where $k$ represents the data with $k$ sets of observations.

If we define

$$
\boldsymbol{\Phi}_k = \begin{bmatrix} \phi_1^T \\ \vdots \\ \phi_k^T \end{bmatrix} \in \mathbb{R}^{k \times n}, \mathbf{Y}_k = \begin{bmatrix} y_1 \\ \vdots \\ y_k \end{bmatrix} \in \mathbb{R}^{k \times 1}
$$

then the algorithm of least squares is

$$
\hat{\boldsymbol{\theta}}_k = \left( \boldsymbol{\Phi}_k^T \boldsymbol{\Phi}_k \right)^{-1} \boldsymbol{\Phi}_k^T \mathbf{Y}_k
\tag{9}
$$

If the data are continuously transferred online, it is quite resource and memory consuming to keep using least squares for solving, so a recursive form is needed to ensure the online updating of $\hat{\boldsymbol{\theta}}_k$. The formula for $\hat{\boldsymbol{\theta}}_k$ is provided in the literature [34]

$$
\begin{aligned}
\varepsilon_k &= y_k - \phi_k^T \hat{\boldsymbol{\theta}}_{k-1} \\
\hat{\boldsymbol{\theta}}_k &= \hat{\boldsymbol{\theta}}_{k-1} + K_k \varepsilon_k \\
K_k &= \mathbf{P}_k \phi_k \\
\mathbf{P}_k &= \mathbf{P}_{k-1} - \frac{\mathbf{P}_{k-1} \phi_k \phi_k^T \mathbf{P}_{k-1}}{1 + \phi_k^T \mathbf{P}_{k-1} \phi_k}
\end{aligned}
\tag{10}
$$

In this paper, Equation (10) is used for online system identification. The state space expression for VCSC was derived in the previous section. For online system identification, the state space equation of the system is rewritten as

$$
\begin{aligned}
\dot{x}_1 &= x_2 \\
\dot{x}_2 &= -\theta_{21} x_1 - \theta_{22} x_2 - \theta_{23} F_L + \theta_{24} x_3 \\
\dot{x}_3 &= -\theta_{31} x_2 - \theta_{32} x_3 + \theta_{33} \sqrt{|p_s - \tanh(k_a x_4) x_3|} x_4 \\
\dot{x}_4 &= -\theta_{41} x_4 + \theta_{42} u
\end{aligned}
\tag{11}
$$

where

$$
\begin{aligned}
&\theta_{21} = \frac{k_c}{m}, \theta_{22} = \frac{b_c}{m}, \theta_{23} = \frac{1}{m}, \theta_{24} = \frac{A_c}{m} \\
&\theta_{31} = \frac{4\beta_e A_c}{V_t}, \theta_{32} = \frac{4\beta_e C_l}{V_t}, \theta_{33} = \frac{4\beta_e}{V_t} C_d w \sqrt{\frac{1}{\rho}} \\
&\theta_{41} = \frac{1}{\tau_v}, \theta_{42} = \frac{k_v}{\tau_v}
\end{aligned}
\tag{12}
$$

We assume that we do not know the parameters of the dynamic characteristics of the servo valve; therefore, parameter identification is conducted for the whole system, and the matrix form of the state space equations of the system that has to be identified is

$$
\begin{bmatrix} \dot{x}_2 \\ \dot{x}_3 \\ \dot{x}_4 \end{bmatrix} = \begin{bmatrix} -x_1 & -x_2 & -F_L & x_3 & 0 & 0 & 0 & 0 & 0 \\ 0 & 0 & 0 & 0 & -x_2 & -x_3 & \sqrt{|p_s - \tanh(k_a x_4)x_3|}x_4 & 0 & 0 \\ 0 & 0 & 0 & 0 & 0 & 0 & 0 & -x_4 & u \end{bmatrix} \begin{bmatrix} \theta_{21} \\ \theta_{22} \\ \theta_{23} \\ \theta_{24} \\ \theta_{31} \\ \theta_{32} \\ \theta_{33} \\ \theta_{41} \\ \theta_{42} \end{bmatrix}
$$

The above dynamic characteristic equation is rewritten as

$$
\mathbf{y} = \mathbf{\Psi}^T \mathbf{\Theta} \tag{13}
$$

where

$$
\mathbf{y} = \begin{bmatrix} \dot{x}_2 & \dot{x}_3 & \dot{x}_4 \end{bmatrix}^T
$$

$$
\mathbf{\Psi}^T = \begin{bmatrix} -x_1 & -x_2 & -F_L & x_3 & 0 & 0 & 0 & 0 & 0 \\ 0 & 0 & 0 & 0 & -x_2 & -x_3 & \sqrt{|p_s - \tanh(k_a x_4)x_3|}x_4 & 0 & 0 \\ 0 & 0 & 0 & 0 & 0 & 0 & 0 & -x_4 & u \end{bmatrix}
$$

$$
\mathbf{\Theta} = \begin{bmatrix} \theta_{21} & \theta_{22} & \theta_{23} & \theta_{24} & \theta_{31} & \theta_{32} & \theta_{33} & \theta_{41} & \theta_{42} \end{bmatrix}
$$

The corresponding online identification algorithm can be developed based on Equations (10) and (13).

## 4. Nonlinear Backstepping Control Method

As can be seen from Equation (7), the state space expression is a nonlinear system, and the backstepping method is needed to obtain a better control effect. The basic principle of the backstepping method is to construct a virtual control quantity equal to the order according to the order of the state space equation, that is, to construct feedback linearization for each subequation of the state space equation such that the nonlinear equation becomes linear and closed-loop control is performed. If more accurate parameter values of system state space expression can be obtained, the control effect of backstepping method is better than that of a traditional control strategy, such as PID.

When the servo valve is considered as the first inertia link, the state space expression of the SCSC system is the fourth order, so the backstepping control algorithm should be carried out in four steps.

The first step of backstepping. Set $i = 1$, define

$$
e_1 = x_1 - x_{1,d}
$$

Take the derivative of the above equation, then

$$
\begin{aligned} \dot{e}_1 &= \dot{x}_1 - \dot{x}_{1,d} \\ &= x_2 - \dot{x}_{1,d} \end{aligned}
$$

Define a Lyapunov function as

$$
V_1 = e_1^2 / 2
$$

and the derivative of the Lyapunov function is

$$
\dot{V}_1 = e_1 \dot{e}_1 = e_1 (x_2 - \dot{x}_{1,d})
$$

when

$$
x_2 = x_{2,d} = \dot{x}_{1,d} - k_1 e_1
$$

The derivative of the Lyapunov function is less than 0, i.e.,

$$\dot{V}_1 = e_1 \dot{e}_1 = e_1(x_2 - \dot{x}_{1,d}) = -k_1 e_1^2 < 0$$

The first equation of state space is stable.

The second step of backstepping. Set $i = 2$, define

$$e_2 = x_2 - x_{2,d}$$

Take the derivative of the above equation, then

$$\dot{e}_2 = \dot{x}_2 - \dot{x}_{2,d}$$
$$= -\theta_1 x_2 - \theta_2 F_L + \theta_3 x_3 - \dot{x}_{2,d}$$

Define Lyapunov function as

$$V_2 = V_1 + e_2^2/2$$

The derivative of the above Lyapunov function is

$$\dot{V}_2 = \dot{V}_1 + e_2 \dot{e}_2$$
$$= \dot{V}_1 + e_2[-\theta_1 x_2 - \theta_2 F_L + \theta_3 x_3 - \dot{x}_{2,d}]$$

if

$$x_3 = x_{3,d} = \frac{-k_2 e_2 + \dot{x}_{2,d} + \theta_1 x_2 + \theta_2 F_L}{\theta_3}$$

then

$$\dot{V}_2 = \dot{V}_1 + e_2 \dot{e}_2$$
$$= \dot{V}_1 + e_2[-\theta_1 x_2 - \theta_2 F_L + \theta_3 x_3 - \dot{x}_{2,d}]$$
$$= -k_1 e_1^2 - k_2 e_2^2 < 0$$

The first and second equation of state space are stable.

The third step of backstepping. Set $i = 3$, define

$$e_3 = x_3 - x_{3,d}$$

The derivative of the above equation is

$$\dot{e}_3 = \dot{x}_3 - \dot{x}_{3,d}$$
$$= -\theta_4 x_2 - \theta_5 x_3 + \theta_6 \sqrt{|p_s - \tanh(k_a x_4) x_3|} x_4 - \dot{x}_{3,d}$$

Define the third Lyapunov function as

$$V_3 = V_2 + e_3^2/2$$

The derivative of above Lyapunov function is

$$\dot{V}_3 = \dot{V}_2 + e_3 \dot{e}_3$$
$$= \dot{V}_2 + e_3\left[-\theta_4 x_2 - \theta_5 x_3 + \theta_6 \sqrt{|p_s - \text{sgn}(x_4) x_3|} x_4 - \dot{x}_{3,d}\right]$$

if

$$x_4 = x_{4,d} = \frac{-k_3 e_3 + \dot{x}_{3,d} + \theta_4 x_2 + \theta_5 x_3}{\theta_6 \sqrt{|p_s - \text{sgn}(x_4) x_3|}}$$

then

$$\dot{V}_3 = \dot{V}_2 + e_3 \dot{e}_3$$
$$= \dot{V}_2 + e_3 \left[ -\theta_4 x_2 - \theta_5 x_3 + \theta_6 \sqrt{|p_s - \text{sgn}(x_4) x_3|} x_4 - \dot{x}_{3,d} \right]$$
$$= -k_1 e_1^2 - k_2 e_2^2 - k_3 e_3^2 < 0$$

Then, the closed-loop control of the first three equations in the state space equation is stable.

In the fourth step, define

$$e_4 = x_4 - x_{4d}$$

The derivative of equation is

$$\dot{e}_4 = \dot{x}_4 - \dot{x}_{4d}$$
$$= -\theta_6 x_4 + \theta_7 u - \dot{x}_{4d}$$

Define the forth Lyapunov function as

$$V_4 = V_3 + \frac{1}{2} e_4^2$$

The derivative equation of the above function is

$$\dot{V}_4 = \dot{V}_3 + e_4 \dot{e}_4$$
$$= \dot{V}_3 + e_4 (-\theta_7 x_4 + \theta_8 u - \dot{x}_{4d})$$

If the input signal of the servo valve is defined as

$$u = \frac{-k_4 e_4 + \dot{x}_{4d} + \theta_7 x_4}{\theta_8}$$

then

$$\dot{V}_4 = \dot{V}_3 + e_4 \dot{e}_4$$
$$= \dot{V}_3 + e_4 (-\theta_7 x_4 + \theta_8 u - \dot{x}_{4d})$$
$$= -k_1 e_1^2 - k_2 e_2^2 - k_3 e_3^2 - k_4 e_4^2 < 0$$

The closed-loop state space equation of the whole system is stable.

## 5. Hardware-In-the-Loop Simulation Verification

In order to verify the correctness of the above online identification algorithm and the backstepping method, the corresponding program needs to be written and ported to the physical controller for verification. However, if the physical controller is directly connected to the entity of the controlled object, the direct control of the physical object is likely to cause unpredictable results, since the control algorithm is still in the debugging stage, which may result in damage to the controlled object or even cause personal injury in serious cases. The traditional solution is to simulate and verify the identification or control algorithm in a simulation environment and then, after this is successfully carried out, to transfer it to the controller to control the physical entity. The simulation environment is nonetheless different from the physical entity, and the algorithm, though successfully debugged in the simulation, may cause various unpredictable problems when transplanted to the physical controller. It would be very advantageous for control system engineers to be able directly debug the algorithm program of the physical controller in a relatively safe environment. HIL simulation technology is created to solve the above problems.

In this paper, the correctness of the online identification and the nonlinear control algorithm is verified using the HIL simulation technique. The HIL simulation technique replaces the physical entity of the controlled object with a mathematical model and operates together with a physical controller in a real-time environment. The physical controller is connected to the real-time solution of the controlled object through the I/O interface of the data acquisition card as if the controller were directly connected to the sensors of the

physical system. Using the above method, it is not aware of the existence of the simulation model, as if it were running together with the physical entity of the controlled object in a real environment. More importantly, in this environment, the HIL simulation algorithm and the backstepping algorithm can be fully debugged because the controlled object is not a physical entity, and there is no need to worry about possible physical damage or personal injury caused by algorithm errors. At the same time, the successfully debugged physical controller can be directly connected to the physical entity of the controlled object, and the control of the physical entity of the controlled object can be basically and immediately successful based on HIL simulation in the early stage, thus improving the development efficiency and reducing the cost.

To improve the reliability of HIL simulation, two aspects need to be addressed.

1.　The mathematical model of the physical entity that replaces the controlled object should be as accurate as possible.
2.　The operating environment of the HIL simulation should be as real time as possible.

Point 1 guarantees the credibility of HIL simulation. If the mathematical model of the controlled object cannot accurately simulate the dynamic performance of the real physical entity, it deviates from the original intention of using HIL simulation. Point 2 ensures the reliability of HIL simulation, because only in a strongly real-time environment can the dynamic characteristics of the controlled physical entity be accurately reproduced, the real-time performance of the control algorithm guaranteed, and the reliability of controller verified.

Regarding point 1, the dynamic equation of the VCSC, as shown in (7), has been described in the previous section. Using this form, the requirements of the system dynamic characteristic equations for both online identification and backstepping simulation can be satisfied. The system parameters obtained from the identification can be directly adopted by the backstepping control algorithm, which ensures the coordination of the online identification algorithm and the backstepping control algorithm. Meanwhile, the equation of (7) takes into account the load spring stiffness, the internal leakage of the hydraulic cylinder, etc., and can reproduce all the dynamic characteristics of the controlled physical entity, and this characteristic equation is accurate.

Regarding point 2, considering the economy and feasibility, the solution adopted in this paper is to install the Preemp_RT patch to the kernel of Linux operating system to ensure the Linux is a hard real-time operating system. The most important feature of the hard real-time kernel is its preempt ability, where ready high-priority tasks are able to preempt low-priority tasks. This ensures the accuracy in timing of the real-time system. After actual testing, the timing accuracy of the Linux system with the Preempt_RT kernel installed can reach 1 ms and is very stable, thus fully meeting the requirements of this paper for system identification and backstepping control algorithm verification.

The validation of the HIL simulation consists of three main groups of experiments.

The first group of experiments verifies the performance of the HIL simulation environment, and the scheme used is the test comparison of 1 ms timer timing accuracy under a Windows environment and 1 ms timer timing accuracy after the Linux+Preempt_RT real-time kernel patch.

In the second group of experiments, the accuracy and reliability of system identification are verified under the HIL simulation environment.

In the third set of experiments, the accuracy and reliability of the backstepping algorithm are verified under the HIL simulation environment.

*5.1. HIL Simulation Platform Hardware and Software*

In order to verify the system identification and backstepping control method in a HIL simulation environment, this paper modifies the Ubuntu 18.04 system into a hard real-time by installing the Preempt_RT patch with the help of Ubuntu 18.04. A low-cost HIL simulation software and hardware platform is established through C language and development of the related software.

The physical hardware of the platform includes an industrial control computer with Ubuntu 18.04 system installed. There is one Advantech USB-5820 analog output data acquisition card and two USB-5817 input data cards. As the Linux operating system and its real-time kernel patch Preempt_RT are open source software, the procurement costs of Advantech data acquisition cards are relatively low under the premise of ensuring 16-bit acquisition accuracy, so the cost of entire HIL simulation platform is relatively low is comparison to commercial products.

The operation principle of the HIL simulation platform is that, with the help of the open source numerical algorithm library GSL [50], the system of differential equations described by the state space is numerically solved using the Runge–Kutta algorithm under the Linux+Preempt_RT real-time environment, and the results of solution operation are outputted by the USB-5820 data acquisition card for analog output (usually ±10 V or 4–20 mA signal). In this way, the physical object under control and its sensor output are simulated, providing a real-time input signal for the controller. At the same time, the HIL simulation platform, with the help of the USB-5817 analog input data acquisition card, reads the control signal of the physical controller (usually ±10 V or 4–20 mA signal) with a period of 1 ms, which is applied to the input of the state space equation, and the differential equation is solved and calculated based on the new control input to obtain the new state variable, which is then outputted with the help of USB-5820 data acquisition card. Outputting continues, so on and so forth, and the basic function of the HIL simulation platform is completed.

The hardware architecture of the HIL simulation platform developed in this paper is shown in Figure 2.

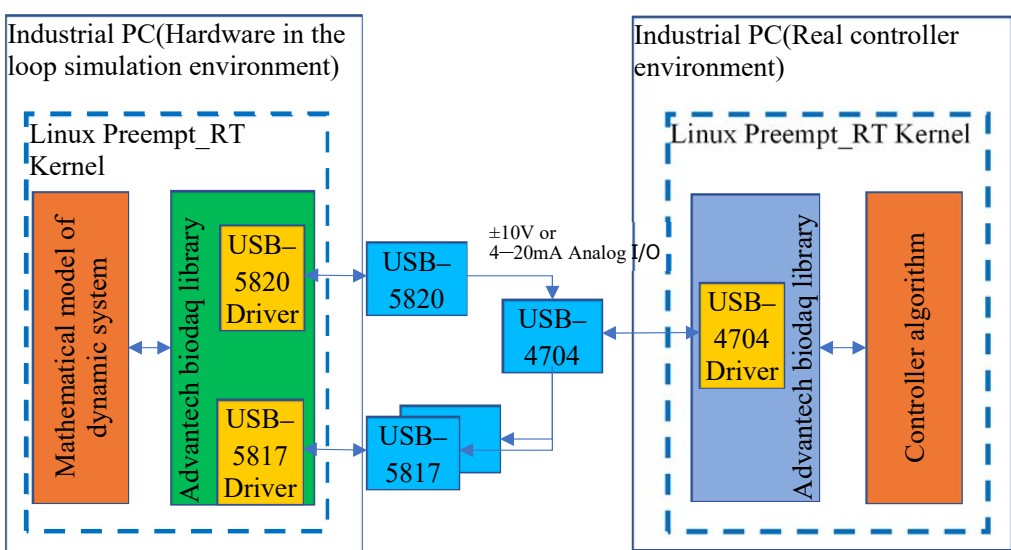

**Figure 2.** Hardware architecture schematic of the hardware-in-the-loop simulation platform.

The HIL simulation platform on which the experiments in this paper are conducted is shown in Figure 3.

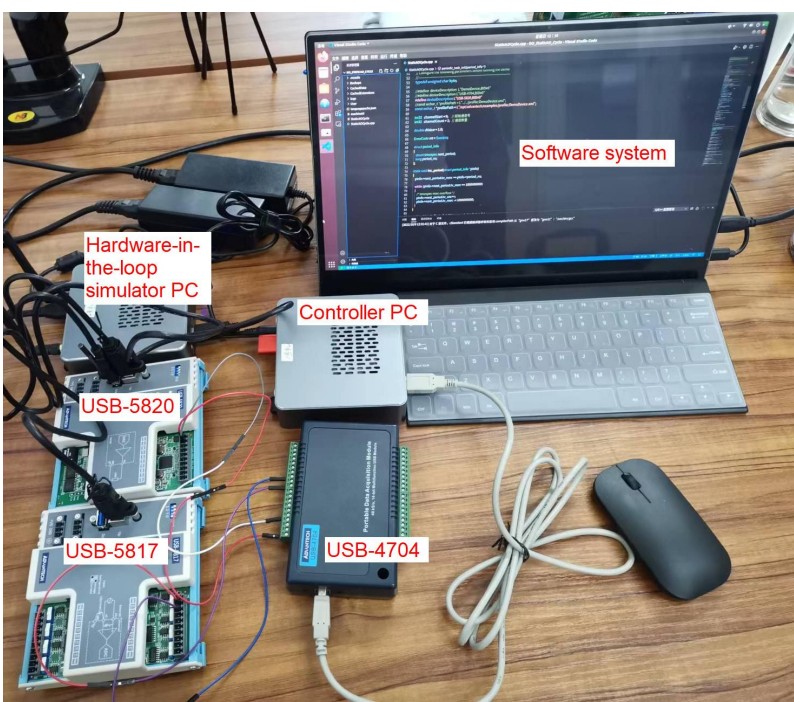

**Figure 3.** Photo of the HIL simulation platform as a physical entity.

The development of software for this HIL simulation platform is realized using C language and calling library functions of Advantech data acquisition cards. Since Advantech data acquisition cards provide drivers for the Linux environment, it ensures the reading of high-precision control signals and the output of numerical calculation results in a real-time environment.

The software operation flow chart of the HIL simulation platform is shown in Figure 4.

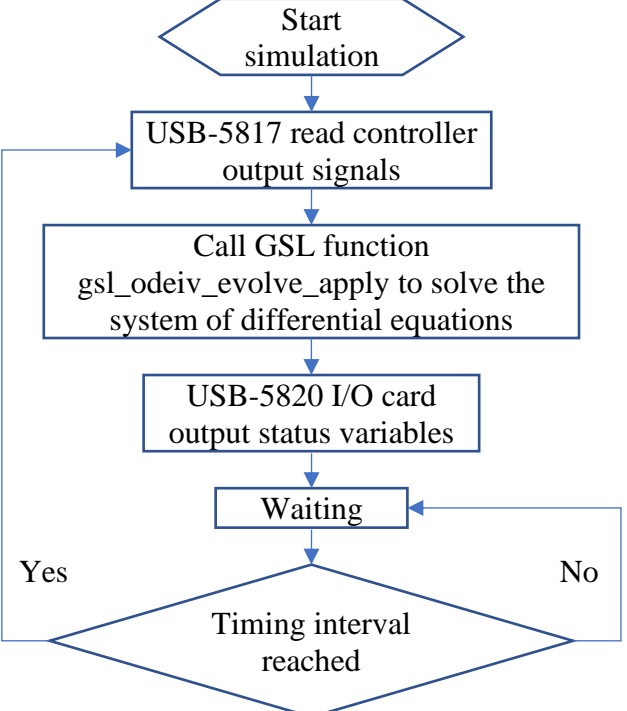

**Figure 4.** HIL simulation platform software operation flow chart.

To complete the HIL simulation experiment, cooperation of the physical controller is also required. In the latter, the cycle of data acquisition, operation, and output control signal is also executed with a timing cycle of 1 ms. The program flow chart is similar to Figure 4, except that the numerical solution part of the differential equation is replaced with either the system identification or the backstepping algorithm, depending on the content of control experiment. The former is used to estimate the key parameters of the system, and the latter is used to perform nonlinear control of the system. The execution process of the algorithm in the physical controller is divided into four main steps.

Step 1: Online identification, calculate the nonlinear coefficients of the electro-hydraulic cylinder system;

Step 2: Update the control parameters of the nonlinear backstepping algorithm;

Step 3: Calculate the next output $u(k+1)$ of the controller;

Step 4: Return to Step 2 and continue the loop.

The schematic diagram of the VCSC model injected by this hardware in the ring HIL simulation platform is shown in Figure 1. The key parameters of the whole model simulation are listed in Table 1.

**Table 1.** Key parameters of the electro-hydraulic servo system of the valve-controlled symmetrical cylinder.

| Parameter | Value | Unit | Specification |
|-----------|-------|------|---------------|
| $k_c$ | $1 \times 10^5$ | N/m | Load spring stiffness |
| $b_c$ | 10 | N/(m/s) | Viscous damping coefficient |
| $D$ | 0.032 | m | Piston diameter |
| $d$ | 0.020 | m | Rod diameter |
| $g$ | 9.81 | m/s$^2$ | Gravity of acceleration |
| $m$ | 100 | kg | Load mass |
| $F_L$ | 1000 | N | External load force |
| $\beta_e$ | $1.7 \times 10^9$ | Pa | Elastic modulus of oil |
| $C_d$ | 0.62 | | Orifice flow coefficient |
| $d_v$ | 0.01 | m | Servo valve spool diameter |
| $w$ | $d_v\pi = 0.0314$ | m | Valve area gradient |
| $p_s$ | $2.1 \times 10^7$ | Pa | Supply pressure |
| $k_a$ | 1000 | | Gain of hyperbolic tangent function |
| $\rho$ | 890 | kg/m$^3$ | Oil density |
| $\tau_v$ | 0.05 | s | Servo valve time constant |
| $k_v$ | $0.015 \times 10^{-3}$ | m/mA | Flow gain of valve |
| $b$ | $d\pi = 0.0628$ | m | Width of internal leakage gap |
| $\delta$ | $1 \times 10^{-4}$ | m | Height of internal leakage gap |
| $\mu$ | 0.051 | Pa s | Dynamic viscosity of oil |
| $l$ | 0.06 | m | Length of internal leakage |
| $C_l$ | $b\delta^3/(12\,\mu L)$ | | Internal leakage coefficient |

### 5.2. Verification of the HIL Simulation Platform Performance in Real Time

In order to verify the real-time performance of the Ubuntu system after installing the Preempt_RT patch, a test is designed for comparing the performance between the Ubuntu18.04 and Windows 10 system in real time. The basic principle of the experiment is that in Ubuntu18.04 system with the Preempt_RT patch installed, the pulse voltage signal is sent out by the data acquisition card USB-5820 of Advantech at a time interval of 1 ms; under the same conditions in the Windows 10 system, using the high-precision multimedia timer and data acquisition card USB-5820 of Advantech, the pulse voltage signal with a period of 1 ms is also sent out. The periodic pulse voltage signals sent by two groups of software timers are collected by the logic analyzer, which is LA1010 produced by Kingst Company in Qingdao, China.

The logic analyzer sampling results for a 1 ms period pulse signal in the Windows 10 environment are shown in Figure 5.

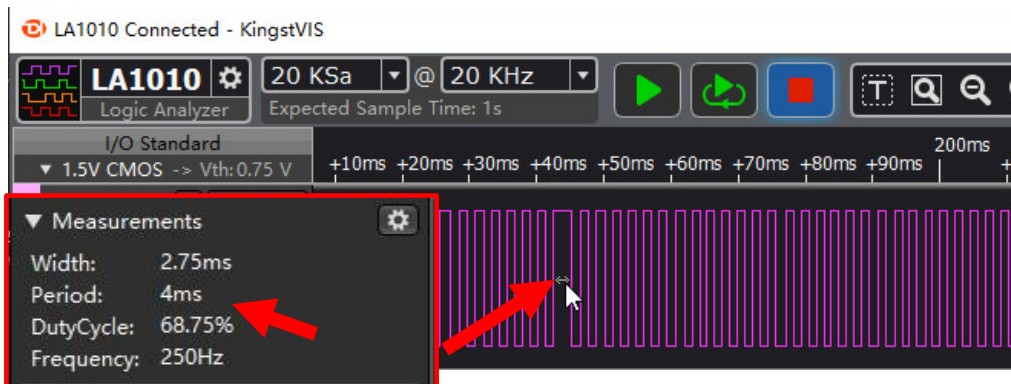

**Figure 5.** Timing accuracy test of high-precision timer under Windows 10 operating system.

It can be observed from the Figure 5 that when the pulse signal is sent at an interval of 1 ms, the period of the pulse signal should have been 2 ms. From the figure, it can be observed that although the vast majority of timing periods are close to 2 ms, there are individual cases of timing periods as long as 4 ms. That is to say, since Windows 10 is not a real-time system, it is not completely reliable to use it for high-precision timing.

In Ubuntu 18.04 with the Preempt_RT patch installed, the voltage signal is sent at 1 ms intervals, and the sampling results obtained with the help of the logic analyzer are shown in Figure 6.

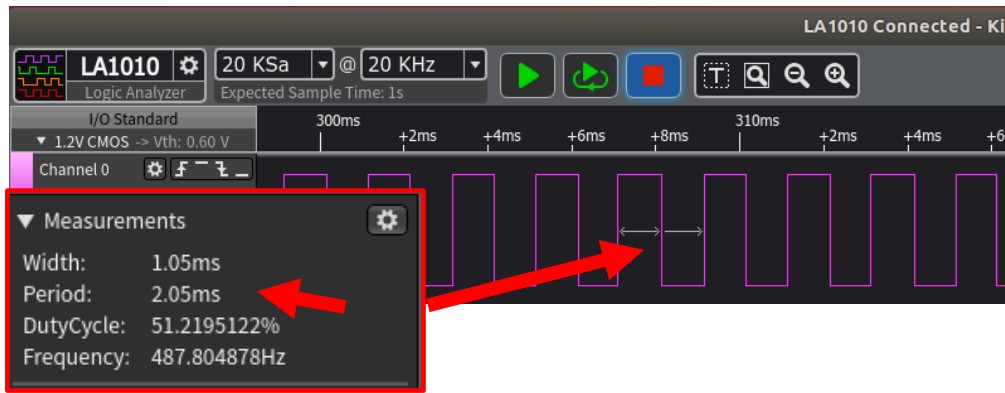

**Figure 6.** Timing accuracy test of high-precision timer under real-time Linux operating system.

From Figure 6, it can be observed that the periodic pulse signal issued under the real-time Linux system is uniform and stable. A timing period is selected for measurement at the maximum in the figure, and the obtained timing period is 2.05 ms, i.e., the maximum timing error is 50 μs.

Comparing Figures 5 and 6, it can be seen that the timing accuracy of the HIL simulation system based on the Linux real-time kernel patch developed in this paper can meet the requirements for real-time sampling and control. The results obtained from the HIL simulation experiments conducted using this platform are therefore credible.

The following paper will use this HIL simulation system to conduct identification and backstepping control experiments.

### 5.3. System Identification Experiments in HIL Simulation Environment

The system identification HIL simulation experiments were conducted on the HIL platform developed in the previous section. The process of system identification is as follows. The HIL simulation platform writes the differential equation of the identified system (valve-controlled symmetric cylinder) in C language on the HIL simulation platform, and the timer time interval is 1 ms. This solution is the output of the identified system, and the output signal is offered to the controller through the USB-5820 data acquisition card.

The controller is another industrial control computer that reads the output of the HIL simulation platform through the analog input function of USB-4704 and outputs a sinusoidal signal to the HIL simulation platform through the analog output function of USB-4704. The timing interval is also 1 ms, and so on and so forth. Between each timing cycle, the system identification algorithm, Equation (10), is invoked to generate the recursive identification results. The identification result is saved to provide the relevant parameters for the next backstepping control algorithm.

In this experiment, the input current signal is $u = 10\sin(2\pi t)\text{mA}$. The input and the displacement signal of the hydraulic cylinder output for the HIL simulation are shown in Figure 7, where the solid line indicates the input current signal and the dashed line indicates the hydraulic cylinder output displacement curve.

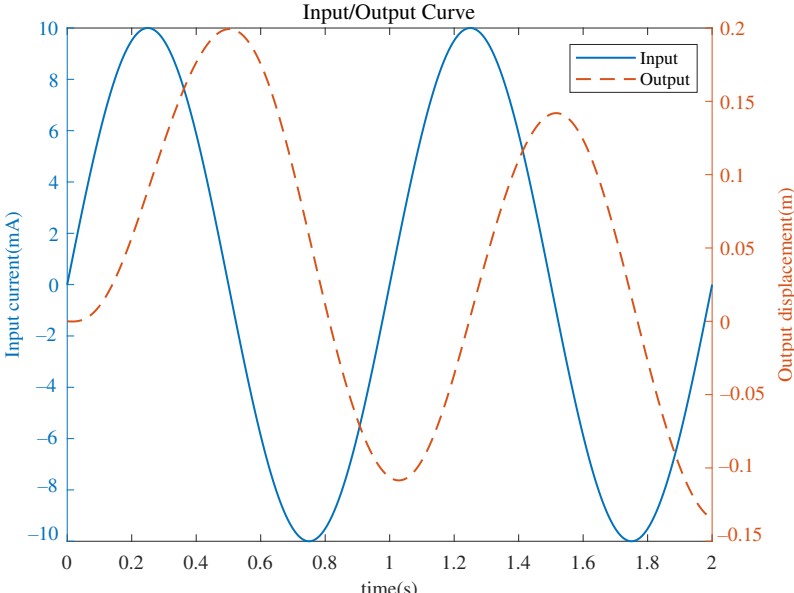

**Figure 7.** Input signal and output displacement curve in the system identification experiment.

Sampling plots of the main state variables and the derivatives of the acceleration and load pressure of the system during identification are shown in Figure 8. The three state variables, velocity, acceleration, and derivative of load pressure, correspond to y in Equation (13), and the values of each element of $\mathbf{\Psi}^T$ in the equation can be calculated from the state variables. Then, according to Equation (10), the value $\mathbf{\Theta}$ of each parameter to be identified can be calculated.

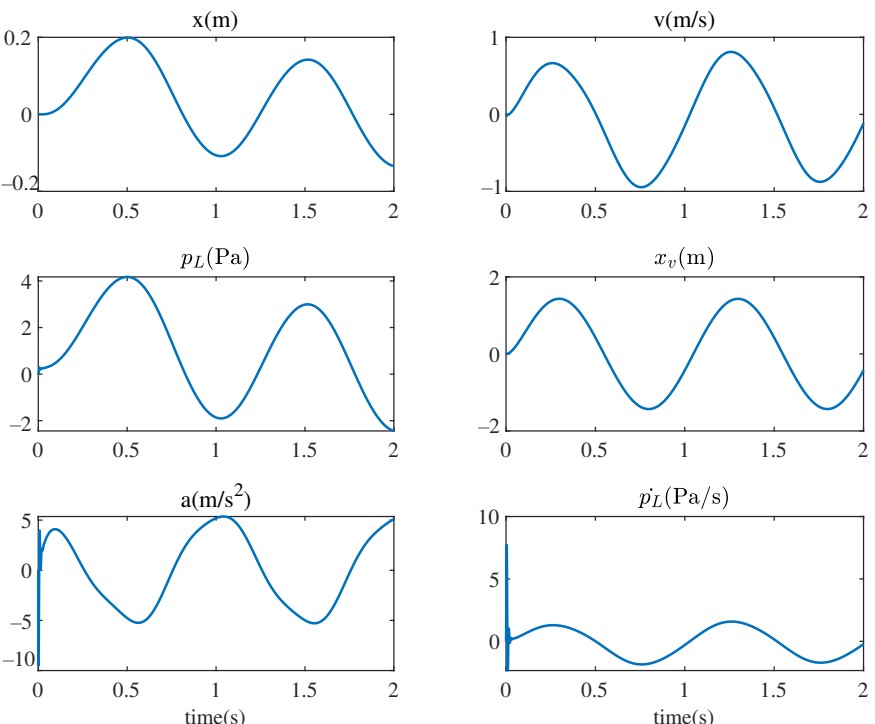

**Figure 8.** Curve diagram of key state variables in system identification experiment.

The iterative approximation curves of the system to identify each parameter are shown in Figure 9. Here, the dashed line is the real value of the identified parameter, which can be obtained according to the HIL simulation model equation. The solid line is the variation curve of the estimated value of the online identification parameter. It can be seen from Figure 9 that, with the accumulation of the number of online recognition iterations, the value of parameter rapidly approaches the true value.

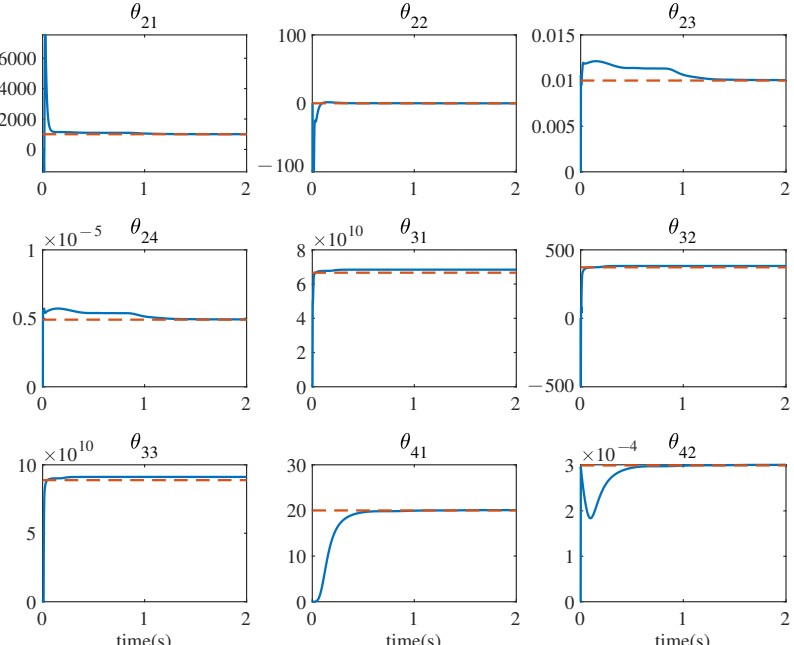

**Figure 9.** Iterative approximation curve of parameters in the system identification experiment.

To further test the correctness of system identification algorithm, a time-varying load is applied to the system with the help of the load simulation function of the platform, i.e., $F_L = [F_{L0} + 1000 \sin(\pi t)]$. Under this time-varying load, the identification results of

the system parameters are still close to the real values, as shown in Figure 10, and the identification results of the parameters are shown in column 5 of Table 2. Therefore, the system identification algorithm developed in this paper is demonstrated to be correct, and the effectiveness of using the semi-physical simulation platform to verify the online identification algorithm is also shown.

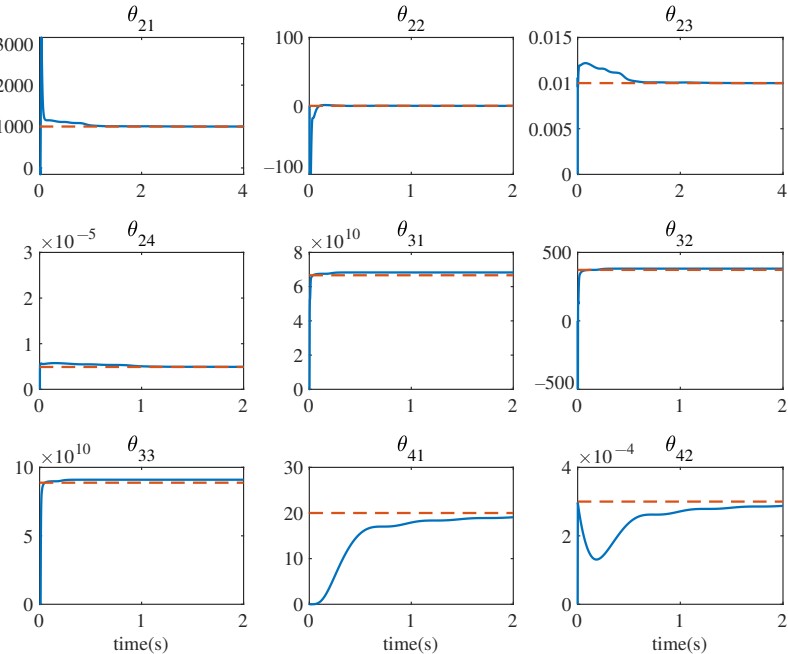

**Figure 10.** Iterative approximation curve of parameters in system identification experiment with time-varying load.

A comparison of the estimated and true values of the stability parameters obtained from system identification is shown in Table 2.

**Table 2.** Comparison between actual and estimated values in system identification of experimental parameters.

| Parameters | Expression | Set Value of Hardware-in-the-Loop Simulation | Estimated Value | Estimated Value of Vary Load |
|---|---|---|---|---|
| $\theta_{21}$ | $k_c/m$ | 1000 | $1.0040 \times 10^3$ | $1.002 \times 10^3$ |
| $\theta_{22}$ | $b_c/m$ | 0.1 | 0.0846 | 0.0801 |
| $\theta_{23}$ | $1/m$ | 0.01 | 0.01 | 0.0100 |
| $\theta_{24}$ | $A_c/m$ | $0.4901 \times 10^{-5}$ | $0.4921 \times 10^{-5}$ | $0.4902 \times 10^{-4}$ |
| $\theta_{31}$ | $4\beta_e A_c/V_t$ | $6.6652 \times 10^{10}$ | $6.2156 \times 10^{10}$ | $6.8303 \times 10^{10}$ |
| $\theta_{32}$ | $4\beta_e C_l/V_t$ | 372.3369 | 347.2963 | 381.5358 |
| $\theta_{33}$ | $4\beta_e C_d w/(V_t\sqrt{\rho})$ | $8.8794 \times 10^{10}$ | $8.2822 \times 10^{10}$ | $9.0988 \times 10^{10}$ |
| $\theta_{41}$ | $1/\tau_v$ | 20 | 20.0821 | 19.6383 |
| $\theta_{42}$ | $k_v/\tau_v$ | $0.3 \times 10^{-3}$ | $0.3014 \times 10^{-3}$ | $0.2953 \times 10^{-3}$ |

As can be seen from Table 2, the values of the parameters obtained from the system identification basically match the set values of the parameters in the HIL simulation environment. This demonstrates the correctness of the online identification algorithm and illustrates the effectiveness of its use in HIL simulation technology.

The parameter values obtained from the system identification are saved and prepared for the performance of backstepping control in the following.

### 5.4. Experiment of Backstepping Control Method in HIL Simulation Environment

Since all the parameters of the state space equation have been obtained based on system identification in the previous section, it is possible to develop the backstepping control program and perform HIL simulation verification of the backstepping control method for the valve-controlled symmetric cylinder system based on the basic principles of Section 3.

The difference from the previous section is that, instead of executing the system identification algorithm in each timing loop of the controller, the backstepping algorithm introduced in Section 3 is executed. The HIL simulation system, however, still uses 1 ms as the timing cycle to complete the fixed cycle of reading the data acquisition card signal, numerically solving the state equation of the VCSC system and outputting the solved state variables through the data acquisition card.

In order to illustrate the effectiveness of the nonlinear control algorithm, HIL simulation of the PID was also performed to verify the effectiveness of the PID control algorithm. The effect of the PID is compared with that of the nonlinear backstepping method.

The HIL simulation model of the VCSC system constructed with the data in Table 1 as key parameters is carried out for the PID and the backstepping control experiment as appropriate. The parameters of the conducted PID experiments are shown in Table 3.

**Table 3.** PID control parameters.

| Control Parameter | Value |
|:---:|:---:|
| $k_p$ | 1000 |
| $k_i$ | 0.01 |
| $k_d$ | 10 |

The graph of simulation results for PID control is shown in Figure 11. The upper part of Figure 11 shows the graph for comparison between the command curve and the actual displacement curve, and the lower part shows the error values of the command curve and the actual output curve.

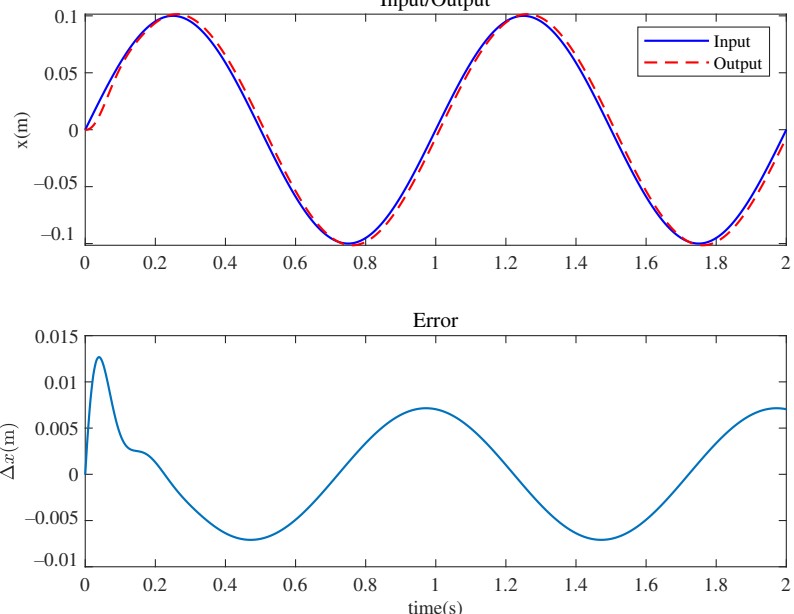

**Figure 11.** HIL simulation curve of PID control.

It can be observed from Figure 11 that the error of the system reaches a maximum of about 0.015 m with the current PID control parameters and then stabilizes at about 0.005 m.

The HIL simulation model of the valve-controlled symmetric cylinder system constructed with the key parameters in Tables 4 and 5 was subjected to two sets of nonlinear backstepping method control HIL simulation experiments. The control parameters of the first group of backstepping method are shown in Table 4.

**Table 4.** The first set of control parameters of the backstepping control method.

| Control Parameter | Value |
| :---: | :---: |
| $k_1$ | 80 |
| $k_2$ | 80 |
| $k_3$ | 160 |
| $k_4$ | 160 |

The first set of hardware-in-the-loop semi-physical simulation experiments for the backstepping method is shown in Figure 12. The command and output curves and the error between them are shown in this figure under the control of backstepping method. As can be seen in the lower part of Figure 12, the system error finally stabilizes within 0.02 m under the action of the control parameters in Table 4, which is approximately the same as the error under PID control.

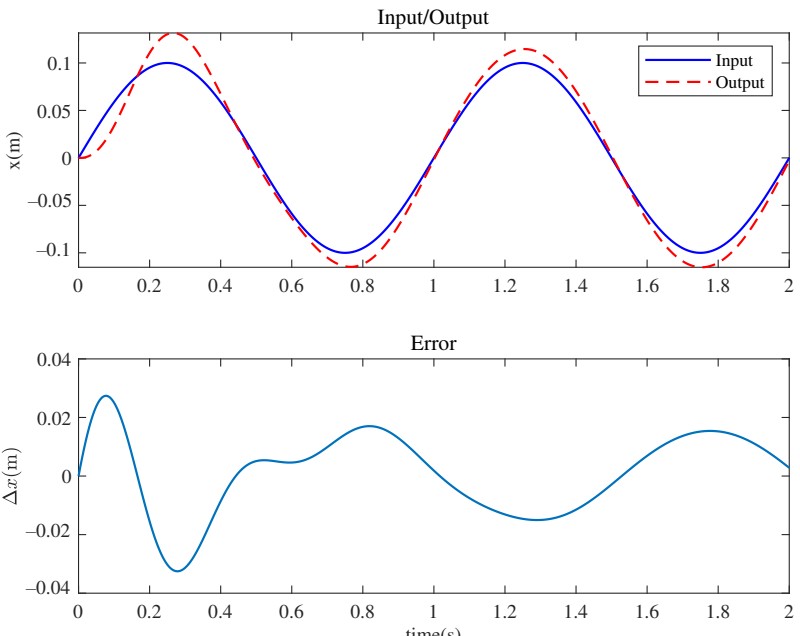

**Figure 12.** HIL simulation curve under the first set of control parameters of backstepping method.

The control parameters for the second set of backstepping methods are shown in Table 5 and have increased compared with the values in Table 4.

**Table 5.** The second set of control parameters of the backstepping control method.

| Control Parameter | Value |
| :---: | :---: |
| $k_1$ | 160 |
| $k_2$ | 160 |
| $k_3$ | 320 |
| $k_4$ | 320 |

The input–output and error curves of the nonlinear backstepping semi-physical simulation of the valve-controlled symmetric cylinder are shown in Figure 13.

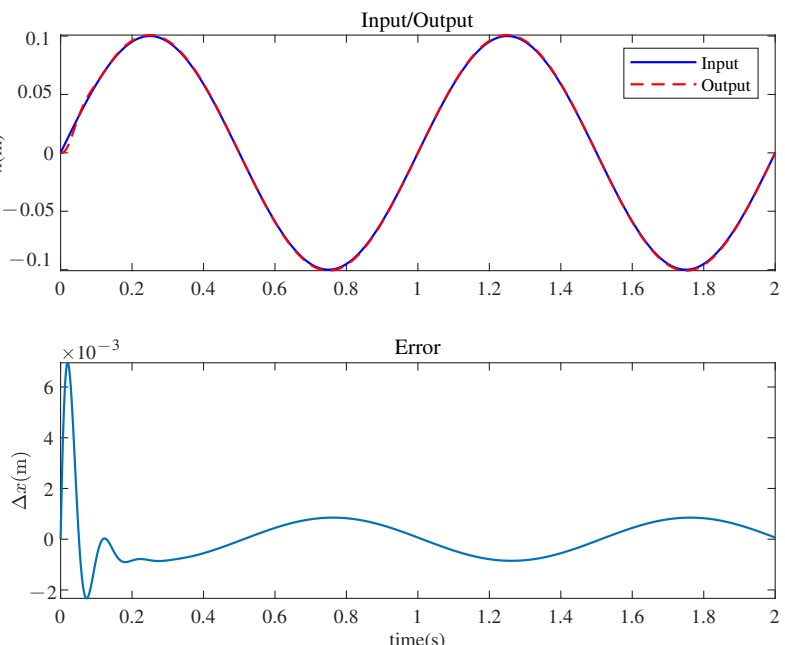

**Figure 13.** Hardware-in-the-loop simulation curve under the second set of control parameters of the backstepping method.

It can be seen from the Figure 13 that, when the values of the control parameters are appropriately increased, the maximum tracking error of the input and output signals is 0.007 m in the initial stage, and as the system operation tends to a steady state, the error rapidly decreases and stabilizes to less than 0.0005 m, which shows that the backstepping method has higher control accuracy compared with the PID control algorithm when the appropriate parameters are set.

## 6. Conclusions

A HIL simulation debugging method based on a Linux real-time kernel patch is proposed to address the problem of difficulty in debugging a EHS system online identification algorithm and nonlinear backstepping control method in practical applications. This developed HIL simulation experiment platform is developed based on Linux+Preempt_RT. Taking the VCSC EHS system as the control object, the online identification algorithm is developed, and the backstepping control algorithm is derived. Using the developed HIL simulation platform, the timer accuracy test is conducted, and the results show that the HIL simulation platform developed in this paper can achieve the required timing accuracy in hard real time. Furthermore, the HIL simulation platform was used to carry out the simulation of system identification and nonlinear backstepping method for VCSC. Using the HIL simulation for system identification, all the key parameters were accurately obtained and assigned to the simulation model of the VCSC in advance; at the same time, using the nonlinear backstepping control implemented by the parameters obtained from system identification, a better control effect was achieved than with the common PID control algorithm.

The above experiments demonstrate that the HIL simulation platform developed in this paper has reliable real-time performance, and the experimental results obtained from the system identification experiments and the nonlinear backstepping control algorithm conducted on this platform are credible. Since the HIL experiments used are simulations, the verification of physical controllers with system identification and control algorithms can be directly applied in engineering the physical entities of VCSCs.

Future research work will focus on building an experimental platform for the physical entity of the VCSC, transplanting the controller verified by the HIL simulation environment to the physical entity experimental platform and completing the system identification and

nonlinear backstepping control experiments for the physical entity to further verify the effectiveness of the system identification and nonlinear backstepping control algorithm.

**Author Contributions:** Conceptualization, Q.L., F.L. and J.G.; methodology, Q.L.; software, K.W.; validation, K.W., H.Z. and Z.W.; formal analysis, D.S.; writing—original draft preparation, Q.L. and J.G.; writing—review and editing, D.S.; visualization, Q.L.; project administration, Q.L.; funding acquisition, D.S. All authors have read and agreed to the published version of the manuscript..

**Funding:** This research was funded by the National Nature Science Foundation of China, grant number 51775354.

**Data Availability Statement:** Not applicable.

**Conflicts of Interest:** The authors declare no conflict of interest.

## Abbreviations

The following abbreviations are used in this manuscript:

| | |
|---|---|
| EHS | electro-hydraulic servo |
| VCSC | valve-controlled symmetrical cylinder |
| HIL | hardware-in-the-loop |

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
