# Peer review of "Application of Hardware-in-the-Loop Simulation Technology in the Development of Electro-Hydraulic Servo System Control Algorithms"

_electronics, doi:10.3390/electronics11233850_

Round 1
Reviewer 1 Report
In the paper the real time identification method of electro-hydraulic servo drive is presented. Also built in Linux real-time Hardware-In-the-Loop environment was used to control of electrohydraulic system. This enabled the verification of the investigated solution. The goal and the topics of the paper sound good. However, there are some issues which should be explained and/or improved:
- In line 122 there is: “kc is the spring stiffness of the external load”, but in my opinion this coefficient should also include the stiffness of the oil in cylinder chambers
- In line 129 is: ” Neglecting the effect of leakage flows” and in line 138 is: “Cl is the coefficient of the total internal leakage of the cylinder”. Please explain
- In line 131 is: “where Vt is the total volume of the cylinder”. Explain if it is total volume in both chambers or only in one of them.
- Explain in more detail: “w is the spool valve area gradient”.
To broaden the overview of the description and modeling of electrohydraulic servo drives, I propose to add the following citations:
H. Murrenhoff, Servohydraulik, Verlag Meinz, Aachen, 1998.
H.E. Merritt, Hydraulic Control Systems, John Wiley & Sons, 1967, ISBN: 978-0-471-59617-2.
A. Milecki, J. Ortmann, Electrohydraulic linear actuator with two stepping motors controlled by overshoot-free algorithm, Mechanical Systems and Signal Processing, vol. 96, November 2017, Pages: 45-57
The paper is focused on identification and control of non-linear system, therefore is not particularly electronics oriented. Therefore Authors should add some issues related to electronics. The main disadvantage of the article is the complete lack of practical verification of the proposed algorithm, i.e. checking the identification and control on a real electrohydraulic drive. This problem should be explained in the article, for example, by showing how parameter changes and disturbances affect the quality of operation of the developed system.
Author Response
- In line 122 there is: “kc is the spring stiffness of the external load”, but in my opinion this coefficient should also include the stiffness of the oil in cylinder chambers
Answer: After checking the literature "Hydraulic Control Systems (Merrit, Herbert E) page 148", kc here really refers to the stiffness of the load spring, excluding the stiffness of the oil in the hydraulic cylinder.
- In line 129 is: ” Neglecting the effect of leakage flows” and in line 138 is: “Cl is the coefficient of the total internal leakage of the cylinder”. Please explain
Answer: This is indeed a clerical error of the author. It should be written as "Do not ignore the rigidity of leakage". It has been corrected in the revised paper.
- In line 131 is: “where Vt is the total volume of the cylinder”. Explain if it is total volume in both chambers or only in one of them.
Answer: The writing here is really not clear enough. This volume should be the total volume of fluid under compression in both chambers.
Answer: W has been explained in the revised draft, w=pi*d, where d represents the diameter of the spool of the cylindrical slide valve.
Answer: I have read and quoted the above literature.
- Explain in more detail: “w is the spool valve area gradient”.
Answer: W has been explained in the revised draft, w=pi*d, where d represents the diameter of the spool of the cylindrical slide valve.
To broaden the overview of the description and modeling of electrohydraulic servo drives, I propose to add the following citations:
- Murrenhoff, Servohydraulik, Verlag Meinz, Aachen, 1998.
H.E. Merritt, Hydraulic Control Systems, John Wiley & Sons, 1967, ISBN: 978-0-471-59617-2.
- Milecki, J. Ortmann, Electrohydraulic linear actuator with two stepping motors controlled by overshoot-free algorithm, Mechanical Systems and Signal Processing, vol. 96, November 2017,Pages: 45-57
Answer: I have read and quoted the above literature in revised draft.
The paper is focused on identification and control of non-linear system, therefore is not particularly electronics oriented. Therefore Authors should add some issues related to electronics. The main disadvantage of the article is the complete lack of practical verification of the proposed algorithm, i.e. checking the identification and control on a real electrohydraulic drive. This problem should be explained in the article, for example, by showing how parameter changes and disturbances affect the quality of operation of the developed system.
Answer: The content of electronics in this paper is the hardware equipment used in the hardware-in-the-loop simulation system. In addition, in order to verify the correctness of the system identification algorithm, the hardware-in-the-loop simulation test of system identification hardware under time-varying external load is supplemented in the revised draft. See lines 447-454 of the revised draft.
Reviewer 2 Report
Review report of electronics-2025285
This paper proposes a method to identify and control an electro-hydraulic servo system in a real-time environment. The method deals with the problem that the parameters of the state space equation of the physical entity of the electro-hydraulic servo system are difficult to obtain accurately. Parameters of the state space model of the valve-controlled symmetrical cylinder system are identified by the recursive least square method. Based on that, a backstepping algorithm is applied to nonlinear control of the valve-controlled symmetrical cylinder Hardware-In-the-Loop simulation platform is built to evaluate the system's performance. Overall, the paper is carefully written and well-structured. The following concerns should be considered:
#1. Please highlight new things proposed in this paper that should be different from others already published, as the reviewer finds that most of the equations listed are not new. Please highlight what the things that you proposed.
#2. The abstract should be revised, and the authors are recommended to list only important things and delete conclusive sentences like: “This identification method can not only obtain the system parameters accurately, but more importantly, the online identification algorithm can improve the efficiency of system identification and adapt to the changes of system structure.”; “With the help of the accurate system parameters obtained by online identification technology in the early stage, the backstepping algorithm can achieve better control effect.”.
#3. Please check if Ref. [2-27] are necessary. And also check if the failure and reliability aspects of the system are helpful and can be considered in this work, such as: 10.1016/j.ress.2022.108777; 10.1016/j.oceaneng.2022.111433; doi.org/10.3390/jmse10111616; 10.1016/j.oceaneng.2021.109261; 10.1016/j.oceaneng.2020.107827.
#4. Another aspect is the validation of the results. Comparing the result of the control method in this paper with existing methods is better.
#5. Except for the above, the paper is well prepared.
Author Response
#1. Please highlight new things proposed in this paper that should be different from others already published, as the reviewer finds that most of the equations listed are not new. Please highlight what the things that you proposed.
Answer: The innovation of the article is reflected in the following aspects:
1) A low-cost hardware-in-the-loop simulation platform based on open source Linux real-time kernel is implemented, which greatly reduces the use cost compared with the commercial hardware-in-the-loop simulation platform;
2) Based on this platform, online identification algorithm and nonlinear backstepping control method are developed. With the help of this platform, the control algorithm can be debugged quickly. At the same time, the development and debugging of this hardware on the hardware-in-the-loop simulation platform will not cause misoperation, damage to equipment or personal injury.
#2. The abstract should be revised, and the authors are recommended to list only important things and delete conclusive sentences like: “This identification method can not only obtain the system parameters accurately, but more importantly, the online identification algorithm can improve the efficiency of system identification and adapt to the changes of system structure.”; “With the help of the accurate system parameters obtained by online identification technology in the early stage, the backstepping algorithm can achieve better control effect.”.
Answer: Thank you for your suggestion. It's already in the revised draft. Delete the corresponding statement.
#3. Please check if Ref. [2-27] are necessary. And also check if the failure and reliability aspects of the system are helpful and can be considered in this work, such as: 10.1016/j.ress.2022.108777; 10.1016/j.oceaneng.2022.111433; doi.org/10.3390/jmse10111616; 10.1016/j.oceaneng.2021.109261; 10.1016/j.oceaneng.2020.107827.
Answer: In order to verify the correctness of the system identification algorithm, the hardware-in-the-loop simulation test of system identification under time-varying external load is supplemented in the revised draft. See lines 447-454 of the revised draft.
#4. Another aspect is the validation of the results. Comparing the result of the control method in this paper with existing methods is better.
Answer: The main innovation of this paper is that the self-built hardware-in-the-loop simulation platform improves the development efficiency of load control algorithm, and eliminates the possible accidents in the traditional development process of control algorithm for physical system.
Reviewer 3 Report
The authors present the Research on Identification and Nonlinear Control of Electro-hydraulic Servo System Based on Hardware-In-the-Loop Simulation. The manuscript is well-organized, including a comprehensive literature review, theory and explanation, experiments and conclusion. Overall speaking, the technique introduced is very interesting and promising. However, the technical contribution is insufficient in the current version and the following points have to be clarified or rectified before the manuscript is suitable for publication:
The authors are expected to clearly define the title of the work.
- Although the results from the proposed method outperformed the corresponding to the other solution methods, it is not clear if the difference can be considered as significant. Is it an important improvement?
- A deep review of the English grammar is required.
- There are many symbols in this paper. 'Notations' should be provided.
Author Response
The authors are expected to clearly define the title of the work.
Answer: After listening to your suggestion, the title of the paper has been revised, and the revised title is: Application of hardware-in-the-loop simulation technology in the development of electro-hydraulic servo system control algorithms.
- Although the results from the proposed method outperformed the corresponding to the other solution methods, it is not clear if the difference can be considered as significant. Is it an important improvement?
Answer: Compared with the previous research, the main improvement of this paper is that the self-built hardware-in-the-loop simulation platform improves the development efficiency of load control algorithm, and eliminates the possible accidents in the traditional development of control algorithm for physical system.
- A deep review of the English grammar is required.
Answer: The English expression of the full text has been corrected again.
- There are many symbols in this paper. 'Notations' should be provided.
Answer: The symbolic explanations of the full text have been listed in formulas and tables.
Round 2
Reviewer 1 Report
The paper can be accepted for publication.
Reviewer 3 Report
The paper is nicely revised. I recommend Accept